# Simple Fabrication of Transparent, Colorless, and Self-Disinfecting Polyethylene Terephthalate Film via Cold Plasma Treatment

**DOI:** 10.3390/nano10050949

**Published:** 2020-05-15

**Authors:** Ji-Hyeon Kim, ChaeWon Mun, Junfei Ma, Sung-Gyu Park, Seunghun Lee, Chang Su Kim

**Affiliations:** 1Advanced Nano-Surface Department, Korea Institute of Materials Science, Changwon 51508, Korea; sangdu87@kims.re.kr (J.-H.K.); apple1025@kims.re.kr (C.M.); ready@kims.re.kr (J.M.); sgpark@kims.re.kr (S.-G.P.); seunghun@kims.re.kr (S.L.); 2School of Architectural, Civil, Environmental, and Energy Engineering, Kyungpook National University, Daegu 41566, Korea

**Keywords:** polyethylene terephthalate, CF_4_ plasma treatment, dry etching, nanopillar structure, self-disinfecting

## Abstract

Cross-infection following cross-contamination is a serious social issue worldwide. Pathogens are normally spread by contact with germ-contaminated surfaces. Accordingly, antibacterial surface technologies are urgently needed and have consequently been actively developed in recent years. Among these technologies, biomimetic nanopatterned surfaces that physically kill adhering bacteria have attracted attraction as an effective technological solution to replace toxic chemical disinfectants (biocides). Herein, we introduce a transparent, colorless, and self-disinfecting polyethylene terephthalate (PET) film that mimics the surface structure of the *Progomphus obscurus* (sanddragon) wing physically killing the attached bacteria. The PET film was partially etched via a 4-min carbon tetrafluoride (CF_4_) plasma treatment. Compared to a flat bare PET film, the plasma-treated film surface exhibited a uniform array structure composed of nanopillars with a 30 nm diameter, 237 nm height, and 75 nm pitch. The plasma-treated PET film showed improvements in optical properties (transmittance and B*) and antibacterial effectiveness over the bare film; the transparency and colorlessness slightly increased, and the antibacterial activity increased from 53.8 to 100% for *Staphylococcus aureus*, and from 0 to 100% for *Escherichia coli*. These results demonstrated the feasibility of the CF_4_ plasma-treated PET film as a potential antibacterial overcoating with good optical properties.

## 1. Introduction

Polyethylene terephthalate (PET) film, often called polyester film, is a versatile plastic used in a variety of applications including packaging (e.g., food, pharmaceutical, health care, medical, industrial, and chemical packaging), electrical (photo-sensitive resistors, insulators, cable and wire wrap, capacitors, circuits), and imaging (X-ray film, instant photos) applications [1]. This is due to its inherent useful qualities such as colorlessness, transparency (clarity), flexibility (or rigidity, depending on the film thickness), flatness, durability, high thermal and electrical insulability, and excellent resilience against impacts, moisture, chemicals, and high temperatures [1]. 

All touchable surfaces (e.g., PET food packaging, PET protective films for mobile devices) used in daily life can cause cross-infections following cross-contamination, a major social concern that has significant implications for human health. Eighty percent of cross infections occur via the following process: people get contaminated by contacting contaminated surfaces by infectees and eventually become infected [2,3]. Accordingly, the development of advanced techniques for antibacterial and antifouling surfaces is urgently needed and has been the subject of significant attention in recent years. Various techniques such as spin-coating, dip-coating, grafting, doctor blading, layer-by-layer, spermine-functionalization, magnetron sputtering, and chemical vapor deposition have been mainly used to fabricate surfaces with antimicrobial, antifouling, and self-cleaning properties [4,5,6].

There are two main antibacterial strategies for surfaces; one includes chemical approaches such as derivatization, functionalization, and polymerization, and the other includes physical approaches in which the surface morphology is modified into nanopatterned (e.g., nanocone, nanopillar, nanoplatelet) structures [7,8,9,10,11,12,13]. Chemical approaches elicit antibacterial effects when the antibacterial components are dissolved and then eluted into bacterial solutions as ions. A wide variety of chemical antibacterial agents including metals, ceramics, polymers, and organic compounds have been developed [4,14,15,16,17]. This chemical approach has several problems to be resolved. Numerous types of bacteria have strong resistance to chemical-based antibacterial agents [18]. Moreover, to elicit a strong antibacterial effect, chemical-based antibacterial agents typically require solutions as an intervening medium. This is because their core elements for killing bacteria are ions released in solutions. When the solution volume is deficient or the solution evaporates immediately after bacterial attachment, the antibacterial activity may be steeply reduced [19]. Physical approaches not only resolve the problems but also have advantages such as short killing-time, long-term effectiveness, and non-toxicity to humans and the environment [20,21,22]. Many researchers have studied the antibacterial effectiveness of nanopatterned surfaces from a bio-physical approach and have suggested that the antibacterial ability of such materials stems from the physical interaction between the surface structures and bacteria. 

Compared to stiff materials (e.g., silicon [7], graphene [8], diamond [9], graphene oxide [11], titanium [12], or titanium oxide [13]), soft polymers have received relatively little study for use in antibacterial nanostructured surfaces through biophysical interactions with the bacterial membrane [19,23,24,25,26,27]. Some studies have found that the antibacterial activity of polyethylene terephthalate (PET) films was enhanced when the physical bactericidal mechanism was activated by modifying the surface structures via cold oxygen plasma [23,24] and nanoimprint lithography [19,25] treatments. Plasma treatment, often used for dry-etching using energetic atoms or molecules, is a useful technique for heat-sensitive materials such as polymers. Furthermore, the application of plasma etching has many advantages, including reproducible and uniform etching results, low operating costs, and simple and environmentally-friendly production. 

This study investigated a carbon tetrafluoride (CF_4_) cold-plasma etching method as a potential technique to impart antibacterial function to transparent PET films with the optical improvement by modifying the surface morphology. The structural and chemical surface properties of bare and plasma-treated PET films were assessed to determine their antibacterial performance. We also analyzed the suitability of the plasma treatment to achieve the desired optical and antibacterial properties.

## 2. Materials and Methods 

### 2.1. Materials and Plasma Treatment 

100 μm-thick PET film (Aiden, Korea), covered with polyethylene protection films on both sides, was investigated in this study. The PET film was cut to a size of 50 mm × 50 mm, and the protection films were completely removed immediately before a plasma treatment. The PET film was subjected to CF_4_ reactive-ion etching (RIE) using 13.56 MHz radio-frequency plasma equipment (LAT, Osan, Korea). The plasma chamber was evacuated to a base pressure of 6.9 × 10^−4^ Torr, and then CF_4_ working gas at a flow rate of 3 standard cubic centimeters per minute (sccm) was introduced to maintain the working pressure at 3.0 × 10^−2^ Torr. Subsequently, a plasma treatment was carried out for 4 min by applying 100 W power. The distance between the specimen (i.e., PET film) and cathode was set at ~7 cm. Prior to the treatment, the chamber had been cleaned through the application of a plasma treatment for 20 min under an oxygen atmosphere of 1.0 × 10^−2^ Torr. Hereafter, the untreated PET film and the plasma-treated PET film are denoted as “bare PET” and “CF_4_-RIE PET,” respectively.

### 2.2. Characterization 

The optical properties of the bare PET and CF_4_-RIE PET were investigated; the transmittance and absorbance were measured via UV-Vis (Cary 5000 UV-Vis-NIR, Agilent, Santa Clara, CA, USA), and the total light transmittance and B* (yellow-blue level) were evaluated using a haze meter (COH 400, Nippon Denshoku, Tokyo, Japan). The surface morphology was examined using a field-emission scanning electron microscope (FE-SEM) (JSM-7610F, JEOL, Tokyo, Japan) operated at a 5-kV voltage and a 10-μA current under high vacuum conditions. The elemental composition was measured using a high-resolution X-ray photoelectron spectrometer (XPS) (K-Alpha, Thermo Fisher, Waltham, MA, USA) for which the spot and step size were fixed at 400 μm and 1.0 eV, respectively. 

### 2.3. Antibacterial Test 

The antibacterial activity of the bare PET and CF_4_-RIE PET were estimated according to Japanese Industrial Standard (JIS) Z 2801 [28]. *Staphylococcus aureus* (*S. aureus*; Korea Collection for Type Cultures, Jeongeup (KCTC), Jeongeup, Korea) and *Escherichia coli* (*E. coli*; KCTC, Jeongeup, Korea), which are representative Gram-positive and Gram-negative bacteria, were chosen as test strains for this study. The inoculum liquid containing the test strains was prepared as follows: the test strain was grown to a sufficient concentration in a 1/500 nutrient broth medium, and then diluted to a density of 2.7 × 10^5^ colony-forming units per milliliter (CFU/mL) through the addition of soybean casein digest lecithin polysorbate (SCDLP) broth. Next, 0.4 mL of the prepared inoculum was spread onto a 40 × 40 mm^2^ test area in the center of the bare PET and CF_4_-RIE-PET samples and the control surface (untreated standard film; 400 POLY-BAG, Seward, Worthing, UK). The test area was covered with a sterilized film 40 mm × 40 mm in size and then placed inside a sterile petri-dish. The petri-dish was incubated for a day at 35 (±1) °C under humid conditions (relative humidity ≥ 90%). Immediately after incubation, the sample and cover film were washed with 10 mL of SCDLP broth medium to extract the remaining viable bacteria. The extracted liquid was diluted with phosphate-buffered saline using the ten-fold serial dilution method to decrease the bacteria density. The quantity of the bacteria contained in the final diluted solution was assessed via the agar-plate count method; 1 mL of the final diluted solution was intermixed with warm agar medium and then sufficiently incubated for a day at 35 (±1) °C in the humid environment. The bacteria colonies in the incubated agar-plate were counted. To meet statistical accuracy criteria for bacterial numbers in a given specimen, samples are usually plated in triplicate and their colony-forming units (CFUs) counted only from plates yielding between 30 and 300 visible colonies. 

According to the JIS Z 2801 [28], the antibacterial activity of a test sample can be expressed in percentage reduction and log reduction of the viable bacteria number with reference to the control. The percent reduction is calculated using [28]
(1)$$R_{p}=\frac{N_{s}-N_{c}}{N_{c}}\times 100$$
where $R_{p}$ is the percent reduction of viable bacteria, $N_{s}$ and $N_{c}$ are the viable-bacteria number after a given contact time (i.e., 24 h in this study) for a test sample and a control, respectively. The $N_{s}$ and $N_{c}$ are obtained according to the following formulas: (2)$$N_{s}=\frac{C_{s}\times D\times V}{A}$$
(3)$$N_{c}=\frac{C_{c}\times D\times V}{A}\;$$
where $C_{s}$ and $C_{c}$ are the number of countable bacteria colonies in the incubated agar plate of the sample and control, respectively, D is the dilution factor ($10^{s}$ where s is the number of times the ten-fold serial dilution), V is the volume of SCDLP broth SCDLP for wash-out in mL, and A is the test area in cm^2^. In this study, V and A are fixed as 10 mL and 16 cm^2^, respectively. In case C ($C_{s}$ and $C_{c}$) is less than 1 (i.e., there is no observable bacteria colony in the incubated agar plate), C is treated as 1. 

The log reduction can be obtained using the following equation [28]:(4)$$R_{l}=\left(U_{t}-U_{o}\right)-(A_{t}-U_{o})=U_{t}-A_{t}$$
where $R_{l}$ is the log reduction of viable bacteria, $U_{o}$ and $U_{t}$ are the average of the logarithm numbers of viable bacteria for the triplicate control samples, immediately after inoculation and the given contact time, respectively, and $A_{t}$ is the average of the logarithm numbers of viable bacteria of the triplicate test-samples after the given contact time.

## 3. Results and Discussion 

Figure 1 demonstrates that the optical properties of the bare PET film were improved via the plasma treatment; the transmittance over the visible-light wavelength region (400–700 nm) and the total light transmittance increased, while the B* [blueness (−)–yellowness (+)] decreased. That is, the PET film became more transparent and colorless. On the other hand, the absorbance over the visible region was maintained at roughly the same level.

Figure 2 shows the XPS spectra calibrated by fixing the C 1s peak at 284.8 eV. When the CF4-RIE PET surface was quantitatively analyzed using survey spectra (Figure 2a), carbon (C), fluorine (F), and oxygen (O) were found to exist at levels of 47.9, 37.8, and 14.3 at.%, respectively. This means that a lot of F-based materials were newly generated on the PET film surface after the plasma treatment, which is well supported by the F 1s and C 1s peaks of Figure 2b,c. The F 1s and C 1s peaks were fitted to optimized Gaussian-Lorentz functions using the peak-fitting program, AVANTAGE (Thermo Fisher Scientific, Waltham, MA, USA). For the C 1s spectra, the sub-peak at 291.2 eV was assigned to carbon fluoride (CF_2_) [29], and for the F 1s spectra, the sub-peaks at 688.1 and 686.5 eV were attributed to absorbed-fluorine atoms (a-F) and CF_2_, respectively [30]. Hence, it is supposed that a-F and CF_2_ were newly generated on the outer surface of the PET film during the CF_4_ plasma treatment. 

Figure 3 presents the morphological changes induced by the plasma treatment; in contrast with the flat surface of the bare PET, the CF_4_-RIE PET had a regular array structure composed of spherical capped nanopillars with 30 nm diameters, 237 nm heights, and 75 nm pitches (the distance between two pillars) as analyzed using the ImageJ software (1.49, Madison, WI, USA). This nano surface structure of the CF_4_-RIE was inspired by an antibacterial biological system, that of the *Progomphus obscurus* (sanddragon) wing [31]. The wing shows a nanopillar array structure consisting of high-aspect-ratio spherically capped nanopillars with an average diameter of 50 nm and an average height of 241 nm [31]. 

The results of the surface investigations (Figure 2 and Figure 3) suggest that the optical improvement effected by the plasma treatment (Figure 1) is mainly due to the nanopillar array structure. In general, rough surfaces reduce transmittances by increasing the absorption capacity for incident light due to light scattering [32]. However, if the size of arrays is smaller than the light wavelength (400–700 nm), sub-wavelength nanoscale arrays allow a gradient of the refractive index in the direction of incident light, increasing transmittance by suppressing light reflection [33]. On the other hand, the B* reduction is thought to be associated with the pitch of the nanopillar structure. A PET film prepared by treating argon plasma for 4 min (Ar-RIE PET, not included in this paper) was similar in morphology to the CF_4_-RIE PET except for the pitch. The B* value of the CF_4_-RIE PET with a 75 nm pitch decreased from 0.61 to 0.50, while that of the Ar-RIE PET with a 35 nm pitch increased to 0.70. Zhao et al. [34] and Purtov et al. [35] revealed a correlation between the color and pitch of pillar array structures. Zhao et al. [34] reported that the color of a silver/chromium/polymethyl methacrylate (PMMA) nanopillar structure with a 70 nm diameter and 150 nm height changed from brown to white (i.e., decreased in B*) with increasing the pitch. 

Based on the standard bacterial plate counting method (Figure 4a), the antibacterial activity of the bare PET and CF_4_-RIE PET was evaluated as presented in Figure 4b,c. As can be seen in Figure 4a, all the CF_4_-RIE PET triplicate samples were free of countable bacterial colony for *S. aureus* and *E. coli*. The actibacterial activity of the CF_4_-RIE PET was calculated according to Equations (1)–(4), assuming that the countable bacteria colony number is one (i.e., C = 1). The antibacterial activity of the CF_4_-RIE PET increased remarkably from 53.8 (±1.0) to 100% for *S. aureus* (Gram-positive bacteria), and from 0 to 100% for *E. coli* (Gram-negative bacteria). More precisely, the antibacterial activity of the CF_4_-RIE PET can be expressed as log reduction (Figure 4c); the log reduction was 4.6 (i.e., 99.997%) for *S. aureus*, and 6.3 (99.99996%) for *E. coli*. These values are the maximum values that can be obtained from the performed antibacterial tests (that is, the $U_{t}$ was 4.6 and 6.3 for *S. aureus* and *E. coli*, respectively.) 

In general, the antibacterial properties of materials correlate with the materials’ surface structure and composition. The CF_4_-RIE PET is thought to elicit an enhanced antibacterial effect thanks to two surface changes; one is a morphological change from a flat to a nanopillar structure, which physically kills the adhering bacteria [36,37,38]; the other is that the fluorinated surface chemically attacks the attached bacteria by releasing fluoride ions [39,40,41,42]. For the PET films, the hydrophobicity was very similar; the contact-angles of the bare PET and CF_4_-RIE PET were 80.7 (±4.99)º and 82.8 (±6.31)º, respectively. It is supposed that the hydrophobicity for the used water-based inoculum had no significant effect on the antibacterial activity. Some studies have reported that surfaces fluorinated via fluorine plasma treatments have slightly higher antibacterial activity than non-treated surfaces [39,42]. However, both the non-fluorinated Ar-RIE PET (not discussed in this paper) and the CF_4_-RIE PET showed lethal antibacterial properties against *S. aureus* and *E. coli*. Thus, the surface morphology is considered to be the main antibacterial factor of the CF_4_-RIE PET. Many studies have reported that natural [31,36,43,44] and bio-mimetic [9,43,45,46,47,48,49] nanopatterned (e.g., nanocone, nanopillar, nanoplatelet) surfaces showed strong antibacterial activity. These morphologies are believed to activate physical or mechanical biocidal actions [36,37,38]. If the tensile stress at contact points exceeds the tensile strength of bacterial cell walls, the bacteria will die due to cell-wall tearing, which is believed to be a mechanism of cell death [50]. The *Progomphus obscurus* (sanddragon) wing, which features a similar structure with the proposed CF_4_-RIE PET, also elicits an antibacterial effect by causing cell-wall rupturing in *Saccharomyces cerevisiae* [31]. According to a bio-physical model for nanopillar structures [36], the thinner the pillars and the greater their pitch, the stronger the antibacterial properties. 

Besides sufficient tensile stress, structural maintenance of the nanopillar structure of the CF_4_-RIE PET upon attachment of the bacteria is required for effective antibacterial activity. The nanopillars could possibly collapse to the ground due to their own weight [51] or the surface adhesive force [52] when considering their softness and high-aspect-ratio and the light bacteria (normally, 1 picogram per bacterium [53]). As can be seen in Figure 3b, the surface adhesive force of the nanopillars seems to be more dominant than the weight. If the ground collapse of a spherical-capped nanopillar is mainly triggered by an adhesion force to the ground, its critical Young’s modulus ($E_{g}^{*}$) can be expressed as [52]
(5)$$E_{g}^{*}=\frac{2^{11/3}3^{3/4}{\left(1-\nu^{2}\right)}^{1/4}h^{3/2}W}{{\left(\pi d\right)}^{5/2}}$$
where ν is the Poisson ratio, h and d are the diameter and height of the pillar, and W is the work of adhesion. By substituting the known ν (0.38 [54]) and W (572 mN/m [55]) of PET, $E_{g}^{*}$ is estimated as 2.14 GPa for the CF_4_-RIE PET nanopillar with h of 237 nm and d of 30 nm, which is slightly lower than the Young’s modulus of bulk PET (2.76 GPa [56]). Given the morphology (Figure 3b) and antibacterial activity (Figure 4b,c), it is supposed that the CF_4_-RIE PET nanopillar has resistance to recoverable deformation, and hence the morphology was sustained without the ground collapse before and after the test-bacteria inoculation. 

Plasma-treated materials exhibit a variety of nanostructures: lamella structures for PET textiles [23,24], flat-capped nanograss (20–80 nm diameter, 0.5 μm height) [43] and sharp nanoneedle (0.5–20 μm) [45] structures for black silicon, and a sharp nanocone structure (10–40 nm tip, 0.4–1.2 μm width, 3–5 μm height) for diamond [9]. Serrano et. al [23] treated PET sutures with oxygen plasma (24 kHz frequency, 80 W applied power, 0.1 mbar working pressure) for 1–20 min. The plasma-treated PET structures exhibited lamella structures separated by elongated voids with 10–200 nm widths and 1–2 μm lengths. However, these PET structures did not show notable improvement in antibacterial activity against *E. coli*. In addition, Orhan [24] also studied the antibacterial performance of a PET fabric treated with cold oxygen plasma (50–100 kHz, 100 W, 1 mbar, 6 cm cathode-to-sample distance). The plasma-treated PET fabric also presented a lamella structure and slight enhancement in antibacterial activity against *S. aureus* and *E. coli*. On the other hand, our CF_4_-RIE PET, with a nanopillar array structure with high-aspect-ratios, showed effective antibacterial performance. Thus, it can be concluded that antibacterial activity is significantly dependent on surface structure, not material. Despite its lower stiffness (Young’s modulus) than that of metals and ceramics, polymers including PET, polyethylene, and polyvinyl chloride may achieve an antibacterial function through modifications to their surface morphology. For example, PET (200 nm in height and pitch) [25] and PMMA (70–215 nm diameter, 200–300 nm height) [26] nanopillar surfaces fabricated using nanoimprint lithography showed lethal action against *Staphylococcus* and coliform bacteria. 

As well as plasma-etching techniques, nanostructuring techniques for antibacterial surfaces have been studied: nanoimprint lithography [25,26], anodization [48], thermal oxidation [57], electron-beam oxidation [58], hydrothermal etching [57], and electrodeposition [46]. Titanium randomly-oriented nanopillar arrays (40.3 nm diameter) fabricated using hydrothermal etching showed high antibacterial activity against *P. aeruginosa* but low antibacterial activity against *S. aureus* [46]. Titanium-alloy nanospikes fabricated using anodization [48] and thermal oxidation [57] were lethal to *S. aureus* and *E. coli*, respectively. Gold nanopillars (50 nm diameter, 100 nm height), nanorings (100–200 nm diameter, 100 nm height), and nanonuggets (100–200 nm diameter, 100 nm height) fabricated using electrodeposition elicited antibacterial actity against *S. aureus* [46]. 

The CF_4_-RIE PET with a unique nanopillar array surface showed notable improvements in optical characteristics (transparency and colorlessness) and antibacterial performance. This indicates that CF_4_ plasma etching is a useful technique to impart antibacterial properties to various contactable PET products used in daily life (e.g., food and medical packaging, display cover films, book covers, sheet protectors, buttons, goggles) while also improving clarity. Moreover, the CF_4_-RIE PET could serve as an effective and affordable film for wide application across touchable surfaces of residential and public facilities, including healthcare centers, schools, voting booths, public transportation, and accommodation. Likewise, antibacterial PET surfaces produced using the CF_4_ plasma treatment may effectively suppress indirect human-to-human infections. 

## 4. Conclusions

In this study, we present a simple technique to impart antibacterial functionality to industrial-quality transparent PET films via a CF_4_ plasma-etching treatment. The PET film subjected to a 4-min plasma treatment exhibited a regular nanopillar array structure with high aspect ratios, and the film surface contained a-F and CF_2_. The unique nanopillar array structure of the film slightly improved its transparency and colorlessness and, furthermore, notably enhanced the film’s antibacterial performance against *S. aureus* and *E. coli* by activating or strengthening the physical biocidal action. Despite their low stiffness, nanopillars with small diameters and long pitches may have subjected the cell walls of the test bacteria to forces that exceeded the walls’ tensile strength. This study’s findings suggest that the proposed plasma treatment could confer strong bacterial resistance to touchable PET surfaces used in a wide range of applications (e.g., food and medical packaging and protective films for mobile phones, television, and monitors) while also improving the films’ optical properties. The use of plasma etching as a dry-etching technique enables rapid and easy processing with high reproducibility. Furthermore, the plasma-treated PET film developed in this study could be applicable for many contactable surfaces in private and public interior spaces as a low-cost self-disinfecting overcoat due to its transparency and flexibility. 

## Figures and Tables

**Figure 1 nanomaterials-10-00949-f001:**
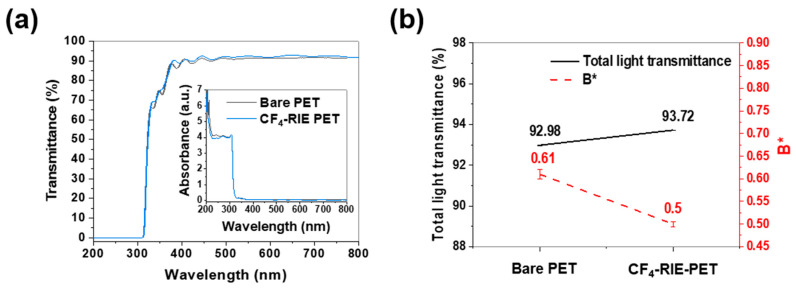
Optical characteristics of bare polyethylene terephthalate film (bare PET) and plasma-treated PET film (CF_4_-RIE PET); (**a**) transmittance and absorbance and (**b**) total light transmittance and B* (yellow-blue level).

**Figure 2 nanomaterials-10-00949-f002:**
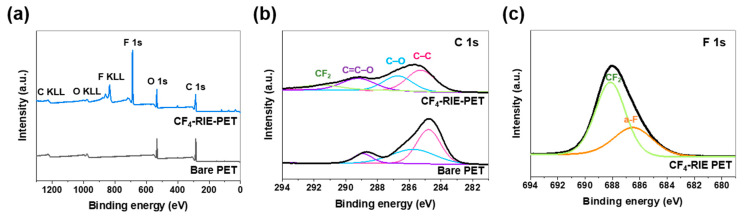
High-resolution XPS spectra of bare PET and CF_4_-RIE PET; (**a**) survey spectra, (**b**) C 1s core-level spectra, and (**c**) F 1s core-level spectra.

**Figure 3 nanomaterials-10-00949-f003:**
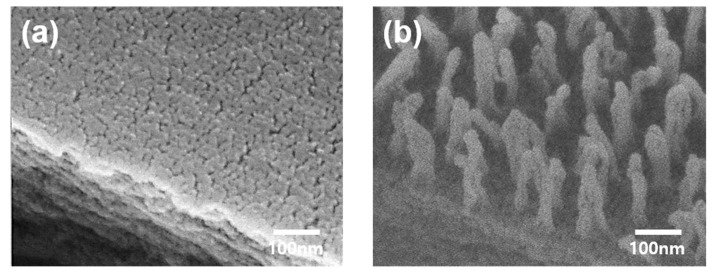
FE-SEM 45°-tilted images of (**a**) bare PET and (**b**) CF_4_-RIE PET.

**Figure 4 nanomaterials-10-00949-f004:**
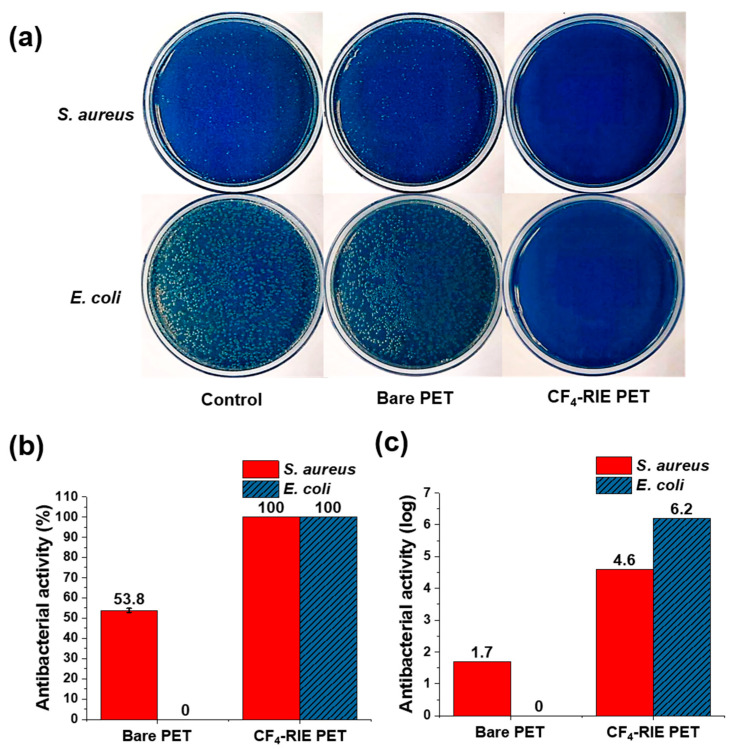
(**a**) Top-view photos of incubated agar plates and (**b**,**c**) antibacterial activity of bare PET and CF_4_-RIE PET. The initial bacteria density on the sample was 1.7 × 10^4^ CFU/cm^2^. The contact time between the sample and bacteria-containing inoculum was 24 h.

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
