# Peer review of "Simple Fabrication of Transparent, Colorless, and Self-Disinfecting Polyethylene Terephthalate Film via Cold Plasma Treatment"

_nanomaterials, 2020, doi:10.3390/nano10050949_

Round 1
Reviewer 1 Report
The research paper "Simple fabrication of transparent, colorless, and self-disinfecting polyethylene terephthalate film via cold plasma treatment” needs a major revision.
1) Authors claimed: “Herein, we introduce a transparent, colorless, and self-disinfecting polyethylene terephthalate (PET) film that mimics the surface structure of the Progomphus obscurus (sanddragon) wing.” What is the relevance of that?
2) The figures appear in the following order: Figure 2, 1, 4 and 3. Please fix that.
3) Authors claimed: “To meet statistical accuracy criteria for bacterial numbers in a given specimen, samples are usually plated in triplicate and their colony-forming units (CFUs) counted only from plates yielding between 30 and 300 visible colonies.” Standard deviations of the results have to be included in the Figure 3B.
4) A graph showing the log reduction values have to be included in the manuscript as well as their standard deviations.
5) The percentage the antibacterial activity increased from 53.8 to 100% (for S. aureus). This result is not very relevant. It means, for instance to have a CFUs reduction from 200 to 100 units (aprox.). Moreover, how is possible that these results led to a 4.6 log reduction. How authors reached to this result? Authors should explain that.
6) Figure 3A is not clear enough at least for S. aureus. Authors should improve the figure for these results.
7) Authors should compare their results with different technologies (such as the use of hydrogels, hot melt extrusion or 3D printing among others) to perform antibacterial surfaces and therefore antibacterial materials that could be used for many applications. Some examples are included below, however, more like these can be found in the literature. Discussion should be improved.
https://doi.org/10.1021/acs.molpharmaceut.6b00402
https://doi.org/10.1016/j.ijbiomac.2019.12.146
https://doi.org/10.2147/IJN.S74811
Author Response
"Please see the attachment."

Reviewer 2 Report
reference 1 does not provide sufficient information for the reader to find the reference. Not sure about its suiability a a generic reference to PET.
It would be better to have journal references for 1 and 2.
What is "mulching vinyl films"
"This nano surface structure of the CF4-RIE was inspired by an antibacterial biological system, system, that of the Progomphus obscurus (sanddragon) wing"
Does not make sense
This nano surface structure of the CF4-RIE resembled that of an antibacterial biological system, ...
The sand dragon wing image should be shown for comparison in Supporting Info.
fluorine ions should be fluoride ions.
Some mention of alternative approaches to forming coatings with anti-microbial, self cleaning surfaces should ould be made in the introduction. For example, polymerization-induced phase segregation Nanomaterials 9 (11), 1610 (2019).
Author Response
"Please see the attachment."

Reviewer 3 Report
The manuscript entitle “Simple fabrication of transparent, colorless, and self-disinfecting polyethylene terephthalate film via cold plasma treatment” is well-described and correct but I consider that it should be extensively improved to be published in Nanomaterials, which has a high factor index.
- Important literature about antimicrobial polymeric systems should be introduced. In example, European Polymer Journal 2015, 65: 46-62; Progress in Polymer Science, 2012, 37, 281-339.
- ml should be mL
- line 1401. “the sub-peaks at 289.3 and 291.2 eV” peaks of F1s are incorrectly assigned.
- Please, define names before using i.e. a-F
- Reorganize the number of Figures, it has no sense that Figure 3 appears before Figure 2.
- Figure 4 should be placed after be named.
- Line 161. 99.9 % (6.3) the reduction is incorrect.
- The contact angle should be performed to see if the hydrophobic character has something to do with the activity.
- How are the mechanical characteristics of material after treatment? These properties are needed to see how the material behaves after modification.
Author Response
"Please see the attachment."

Round 2
Reviewer 1 Report
Accept in present form.
Reviewer 3 Report
The authors have significantly improved the manuscript. Therefore, I consider that it can be now published.
Author Response
Thank you for your comments and suggestions, which have helped us to greatly improve the manuscript.